# Predictors of neonatal mortality in the Eastern Regional Hospital in Ghana: A retrospective cohort study

Diana Awintima Apanga[1]*, Maxwell Tii Kumbeni[2], Abdulai Mohammed Salifu[3], Nana Mireku-Gyimah[4], Paschal Awingura Apanga[5]

1 Paediatric Department, Eastern Regional Hospital, Koforidua, Eastern Region, Ghana, 2 School of Public Health and Nutrition, College of Health, Oregon State University, Corvallis, Oregon, United States of America, 3 Surgical Division, General Surgery Unit, 37 Military Hospital, Accra, Ghana, 4 Mpohor District Health Directorate, Mpohor, Western Region, Ghana, 5 Nuffield Department of Medicine, Jenner Institute, University of Oxford, Oxford, United Kingdom

* dianaapanga@gmail.com

**Data Availability Statement:** All supporting data have been made available in the manuscript.

**Funding:** The authors received no specific funding for this work.

## Abstract

Neonatal mortality accounts for nearly half of under-5 mortality in Ghana. The aim of this study was to identify the predictors of neonatal mortality in the Eastern Regional Hospital, Ghana. This was a retrospective cohort study conducted using secondary data from electronic medical records from the Eastern Regional Hospital between 1st January 2022 and 31st December 2022. The Kaplan-Meier estimator and adjusted Cox regression model were used to estimate survival probability and to assess the predictors of neonatal mortality. Data on 1684 neonates were analyzed and we found that 11.82% deaths occurred with a neonatal mortality rate (NMR) of 13.98 (95% CI: 12.05, 15.91) per 1000 person-days. Most neonatal deaths occurred within the first 24hrs of life (9.9%). The predictors of neonatal mortality were found to be low birthweight [Adjusted hazard rate (aHR): 1.63, 95% CI: 1.04, 2.54], hypothermia (aHR: 1.82, 95% CI: 1.16, 2.85), hyperthermia (aHR: 1.85, 95% CI: 1.01, 3.39), birth asphyxia (aHR: 3.69, 95% CI: 1.68, 8.11), and multiparty (aHR: 1.66, 95% CI: 1.02, 2.70). However, neonates aged 8–28 days (aHR: 0.41, 95% CI: 0.21, 0.81), born in the Eastern Regional Hospital (aHR: 0.39, 95% CI: 0.28, 0.55), walk-in neonates (aHR: 0.54, 95% CI: 0.32, 0.90), and neonates whose mothers had 8 or more antenatal contacts (aHR: 0.54, 95% CI: 0.32, 0.92) had lower neonatal mortality. There was high NMR in the Eastern Regional Hospital in Ghana. Averting complications such as low birthweight, hypothermia, hyperthermia, birth asphyxia, including the provision of obstetric and early neonatal care within the first 24 hours of life is critical to reducing neonatal mortality. Adherence to the World Health Organization's recommendation of 8 or more antenatal contacts among pregnant women is also essential in reducing neonatal mortality.

**Competing interests:** The authors have declared that no competing interests exist.

## Introduction

According to the World Health Organization (WHO), 2.4 million children died globally in the first month of life with an estimated 6700 neonatal deaths occurring daily [1]. Sub-Saharan Africa has a neonatal mortality rate (NMR) of 27 deaths per 1000 live births, which accounts for 43% neonatal deaths globally [1]. The risk of neonatal mortality for a child born in Sub-Saharan Africa is 10 times the risk of neonatal mortality for a child born in a high-income country [1]. Most neonatal deaths in Sub-Saharan Africa occur within the first week of life and are related to complications from the lack of quality care at birth or immediate postpartum. Some of these complications include preterm birth, birth asphyxia, infections, and birth defects [2–5]. Due to the global burden of neonatal mortality, the Sustainable Development Goal (SDG) 3 specifies that all countries should aim at reducing neonatal mortality to at least 12 deaths per 1,000 live births by 2030 [6].

In Ghana, NMR has declined from 43 deaths per 1,000 live births in 1988 to 17 deaths per 1,000 live births in 2022 [7]. Ghana has implemented several policy measures to address neonatal mortality, including the Free Maternal Health Care Policy and the Newborn Health Strategy and Action Plan 2014–2018 [8, 9], which was revised in 2019 [10]. Despite the progress made in Ghana towards reducing NMR, neonatal deaths accounts for 61% of infant deaths and 43% of under-5 mortality [7]. The rate of decline in neonatal mortality is slow and Ghana may not achieve the SDG target of at least 12 deaths per 1,000 live births by 2030 [6], if interventions are not employed. In this regard, understanding the factors associated with neonatal mortality is essential to develop policies and interventions towards achieving SDG 3 in Ghana.

Research on factors associated with neonatal mortality in Ghana has largely been conducted among the general population [11–14], while fewer studies have been conducted in hospital settings [15–18]. Many of these studies did not employ a robust design [15–18]. Literature on predictors of neonatal mortality in the Eastern Region is also lacking. The goal of this study was to assess the predictors of neonatal mortality at the Eastern Regional Hospital in Ghana using a retrospective cohort design. This study focused on neonates admitted in the neonatal intensive care unit (NICU) of the Eastern Regional Hospital. Findings from this study will inform decision and policy making with regards to reducing neonatal mortality in the NICUs in Ghana, particularly in the Eastern Region.

## Methods

### Study design, setting, and population

A retrospective cohort study was conducted in the NICU of the Eastern Regional Hospital, Ghana. The Eastern Regional Hospital is a secondary referral hospital in Eastern Region of Ghana. It serves as a referral point for about 16 district hospitals, including the New Juaben Municipality where it is located. The hospital is readily accessible [19]. The NICU of hospital has a bed capacity of forty (40) with a workforce of two (2) pediatricians and thirty-three (33) nurses. The NICU admits an average of 142 neonates per month. The NICU is resourced with a number of pediatric equipment, but not limited to phototherapy machines, continuous positive airway pressure (CPAP) machines, pulse oximeters, bilirubinometers, glucometers and radiant warmers. The study population comprised of all neonates admitted to the NICU of the Eastern Regional Hospital in Ghana.

### Data source and data collection

We accessed data from the NICU of the Eastern Regional Hospital in Ghana. Data was extracted from electronic medical records of the hospital. Data on neonates admitted to the

NICU between 1st January 2022 and 31st December 2022 were reviewed and collected for this study. Extracted data included information on maternal characteristics (i.e., age, education, marital status, residence, antenatal care, parity, and multiple gestation) and neonatal characteristics (i.e., gestational age, mode of delivery, age, sex, birthweight, referral status, diagnosis, and temperature on admission). The data was extracted between 12th June 2023 and 23rd June 2023.

To estimate the minimum sample size required for this study, a previously reported prevalence of neonatal mortality (20.2%) from a teaching hospital in Ghana was used [16]. Using a 5% margin of error and a 95% confidence interval, an estimated minimum sample size of 248 was required.

## Outcome

The primary outcome was time to death from when the neonate was admitted to the NICU of the Eastern Regional Hospital. The outcome was dichotomized as "1" if the neonate had developed the event (i.e., died) and "0" if the neonate was censored. Censoring in this study occurred if the neonate recovered and was discharged or if the neonate did not experience the event before the study period ended.

## Predictors

Predictor variables of interest in this study included both maternal and neonatal characteristics. Maternal characteristics were categorized as follows; marital status (single, married); mother's education (none, primary, secondary, tertiary); place of residence (rural, urban), number of antenatal contacts (0, 1–7, ≥8); parity (primiparous, multiparous); and multiple gestation (no, yes). Mother's age was assessed as a continuous variable.

Neonatal characteristics were categorized as gestational age (term, preterm), mode of delivery (vaginal delivery, cesarean section), age of neonate (<24 Hours, 24 Hours—<72 Hours, 72 Hours-7 Days, 8 Days-28 Days), sex of neonate (male, female), birthweight (normal weight, low birth weight, macrosomia), and referral status (referred, walk-in, born in the Eastern Regional Hospital). Other neonatal characteristics include temperature (normothermia, hyperthermia, hypothermia) and diagnosis on admission (neonatal sepsis, prematurity, neonatal jaundice, birth asphyxia, pneumonia, other diseases). Other diseases in this study referred to any of the following conditions: meconium aspiration syndrome, meningitis, staphylococcal skin sepsis, congenital heart disease and congenital abnormalities. Predictor variables in our study were selected based on available data in the electronic records from the Eastern Regional Hospital and previous literature [16, 17, 20].

## Data analysis

Descriptive statistics were used to assess characteristics of the study population. Categorical variables were presented with frequencies and percentages, whilst continuous variables were described using mean and standard deviation. The NMR was estimated as the probability of dying during the first 28 days of life expressed per 1,000 live births [1]. We used the Kaplan-Meier method to examine survival of the neonates. Kaplan–Meier curves were plotted for various characteristics of the study population. The equality of the survivor functions (i.e., comparing deaths versus survivors) was assessed using the log-rank test. A P-value of less than 0.05 was considered statistically significant.

We used both univariate and multivariable Cox regression models to assess the predictors of neonatal mortality. Univariate analysis was conducted to characterize the relationship between each predictor variable and neonatal mortality. Predictor variables with a P-value of

less than 0.05 from the univariate analysis were included in the Cox multivariable regression model to identify the predictors of neonatal mortality. We also assessed the Cox multivariable model to ensure that it satisfied the proportional hazard (PH) assumption. The goodness-of-fit (GOF) test, which is defined by Schoenfeld residuals, and the log-log plots were also used to examine the PH assumption. Both descriptive statistics and Cox regression analyses were conducted using SAS version 9.4 (SAS Institute, Cary, NC, USA).

### Ethical approval

This study did not require ethics approval as authors used deidentified secondary data. The data was obtained from a prior primary study, which had ethical approval from the Ghana Health Service Ethical Review Committee (GHS-ERC:061/04/23). Whilst we sought permission from the Eastern Regional Hospital to use the data, patient consent to participate was waived because the data was de-identified.

## Results

### Characteristics of the study population

Data for 1684 neonates were analyzed. A greater proportion of neonates admitted to the NICU were less than 24 hours old (65.1%). Many of the neonates were males (54.3%). Mean age of mothers of neonates in this study was 27.1 years ± 10.7 (Table 1). Other baseline characteristics of the study population are summarized in Table 1.

A total of 199 (11.82%) neonates died and the overall mortality rate was 13.98 (95% CI: 12.05, 15.91) per 1000 person-days. The number of neonates who were diagnosed with prematurity on admission were 73 (4.4%). Most of the deaths, 167 (9.9%) occurred within the first 24 hours after birth (Table 1). Results of the log-rank test showed significant differences in cumulative probability of survival of neonatal characteristics such as gestational age, neonatal age, birthweight, referral status, diagnosis, and temperature on admission (p < 0.05; Fig 1). The survival for neonates with maternal characteristics, including place of residence, number of antenatal contacts and parity were also significantly different between neonates who died and those who survived (p < 0.05; Fig 2).

### Predictors of neonatal mortality

Neonatal characteristics associated with neonatal mortality include age of a neonate, birthweight, referral status, temperature on admission, and diagnosis on admission. The rate of mortality was 59% lower in neonates aged 8–28 days compared to neonates less than a day's (<24 hours) old [Adjusted hazard rate (aHR): 0.41, 95% CI: 0.21, 0.81]. The rate of mortality among low birthweight neonates was 1.63 times the rate among normal weight neonates (aHR: 1.63, 95% CI: 1.04, 2.54). The rate of mortality among neonates who were delivered at the Eastern Regional Hospital (aHR: 0.39, 95% CI: 0.28, 0.55), and neonates whose mothers brought them to hospital on a walk-in basis (aHR: 0.54, 95% CI: 0.32, 0.90), were lower compared to neonates who were referred to the hospital (Table 2). The rate of mortality among neonates with hypothermia and hyperthermia were 1.82 and 1.85 times respectively the rate among neonates with normothermia (Table 2). Whilst birth asphyxia was associated with a higher mortality, neonatal jaundice was associated with a lower mortality (Table 2).

Maternal characteristics such as number of antenatal contacts and parity were associated with neonatal mortality. Neonates whose mothers had 8 or more antenatal contacts had 46% lower hazard for mortality compared to neonates whose mothers did not have contact with an antenatal care provider (aHR: 0.54, 95% CI: 0.32, 0.92). The rate of mortality among neonates

**Table 1. Characteristics of the study population.**

| Variable | Total (n = 1684) | Survivors (n = 1485) | Deaths (n = 199) | Log-rank test, P-value |
|---|---|---|---|---|
| Mother's age, Mean ±SD | 27.1 ± 10.7 | 27.0 ± 10.9 | 27.9 ± 9.1 | 0.225* |
| Marital status, n (%) | | | | |
| Single | 741 (44) | 649 (38.5) | 92 (5.5) | 0.530 |
| Married | 943 (56) | 836 (49.6) | 107 (6.4) | |
| Mother's education, n (%) | | | | |
| None | 260 (15.4) | 223 (13.2) | 37 (2.2) | 0.506 |
| Primary | 205 (12.2) | 179 (10.6) | 26 (1.5) | |
| Secondary | 837 (49.7) | 736 (43.7) | 101 (6.0) | |
| Tertiary | 382 (22.7) | 347 (20.6) | 35 (2.1) | |
| Place of residence, n (%) | | | | |
| Rural | 809 (48) | 688 (40.9) | 121 (7.2) | 0.001 |
| Urban | 875 (52) | 797 (47.3) | 78 (4.6) | |
| Antenatal contacts, n (%) | | | | |
| 0 | 474 (28.1) | 429 (25.5) | 45 (2.7) | 0.001 |
| 1—7 | 712 (42.3) | 599 (35.6) | 113 (6.7) | |
| 8+ | 498 (29.6) | 457 (27.1) | 41 (2.4) | |
| Parity, n (%) | | | | |
| Primiparous | 432 (25.7) | 398 (23.6) | 34 (2.0) | 0.006 |
| Multiparous | 1252 (74.3) | 1087 (64.5) | 165 (9.8) | |
| Multiple gestation, n (%) | | | | |
| No | 1398 (87.4) | 1228 (76.8) | 170 (10.6) | 0.563 |
| Yes | 201 (12.6) | 176 (11.0) | 25 (1.6) | |
| Missing | 85 | | | |
| Gestational age, n (%) | | | | |
| Term | 937 (55.6) | 856 (50.8) | 81 (4.8) | 0.002 |
| Preterm | 747 (44.4) | 629 (37.4) | 118 (7.0) | |
| Mode of delivery, n (%) | | | | |
| Vaginal delivery | 759 (45.1) | 657 (39.0) | 102 (6.1) | 0.076 |
| Cesarean section | 925 (54.9) | 828 (49.2) | 97 (5.8) | |
| Age of neonate, n (%) | | | | |
| <24 Hours | 1096 (65.1) | 929 (55.2) | 167 (9.9) | < .001 |
| 24 Hours—<72 Hours | 184 (10.9) | 169 (10.0) | 15 (0.9) | |
| 72 Hours-7 Days | 120 (7.1) | 115 (6.8) | 5 (0.3) | |
| 8 Days-28 Days | 284 (16.9) | 272 (16.2) | 12 (0.7) | |
| Sex of neonate, n (%) | | | | |
| Male | 915 (54.3) | 810 (48.1) | 105 (6.2) | 0.740 |
| Female | 769 (45.7) | 675 (40.1) | 94 (5.6) | |
| Birthweight, n (%) | | | | |
| Normal weight | 782 (46.4) | 719 (42.7) | 63 (3.7) | < .001 |
| Low birthweight | 702 (41.7) | 578 (34.3) | 124 (7.4) | |
| Macrosomia | 200 (11.9) | 188 (11.2) | 12 (0.7) | |
| Diagnosis on admission, n (%) | | | | |
| Prematurity | 493 (29.8) | 420 (25.4) | 73 (4.4) | < .001 |
| Neonatal Jaundice | 330 (20) | 328 (19.8) | 2 (0.1) | |
| Birth Asphyxia | 59 (3.6) | 38 (2.3) | 21 (1.3) | |
| Pneumonia | 82 (5) | 76 (4.6) | 6 (0.4) | |
| Neonatal Sepsis | 159 (9.6) | 149 (9.0) | 10 (0.6) | |

(*Continued*)

**Table 1.** (Continued)

| Variable | Total (n = 1684) | Survivors (n = 1485) | Deaths (n = 199) | Log-rank test, P-value |
|---|---|---|---|---|
| Other diseases | 531 (32.1) | 469 (28.4) | 62 (3.7) | |
| vMissing | 30 | | | |
| Referral status, n (%) | | | | |
| Referred | 359 (21.3) | 277 (16.4) | 82 (4.9) | < .001 |
| Walk-In | 324 (19.2) | 300 (17.8) | 24 (1.4) | |
| Born in the Eastern Regional Hospital | 1001 (59.4) | 908 (53.9) | 93 (5.5) | |
| Temperature on admission, n (%) | | | | |
| Normothermia | 482 (28.6) | 454 (27) | 28 (1.7) | < .001 |
| Hyperthermia | 203 (12.1) | 180 (10.7) | 23 (1.4) | |
| Hypothermia | 999 (59.3) | 851 (50.5) | 148 (8.8) | |

* is a t-test p-value.

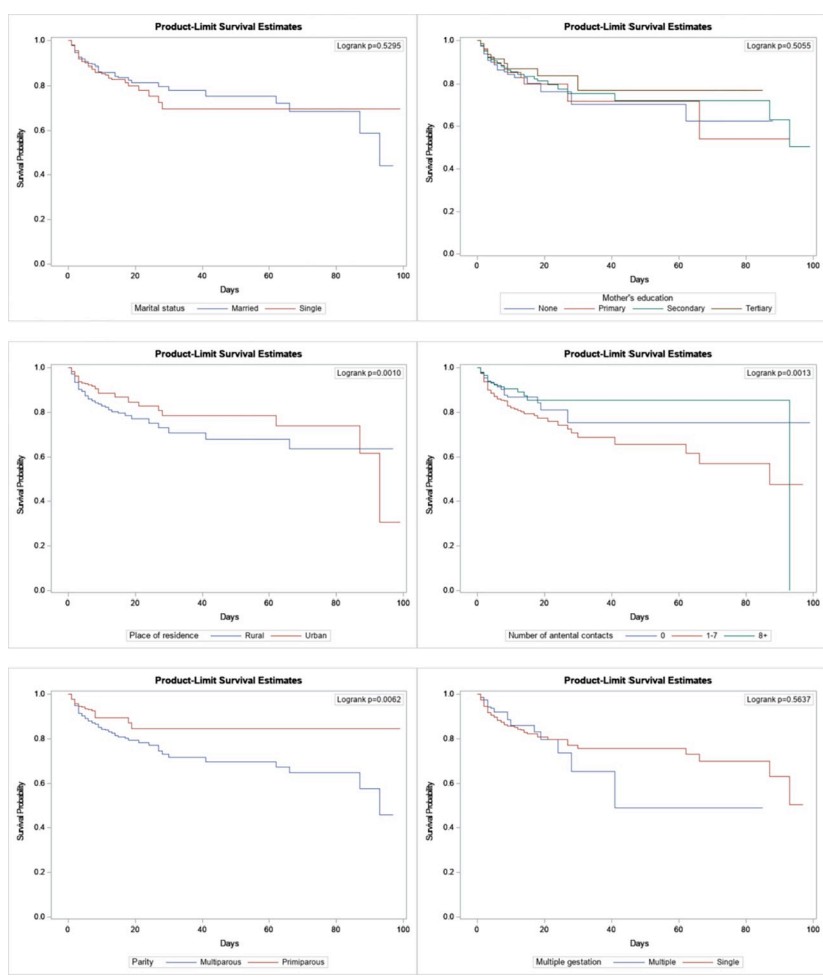

**Fig 1. Kaplan-Meier survival curves of neonates admitted in the NICU of the Eastern Regional Hospital by neonatal characteristics, 2022.**

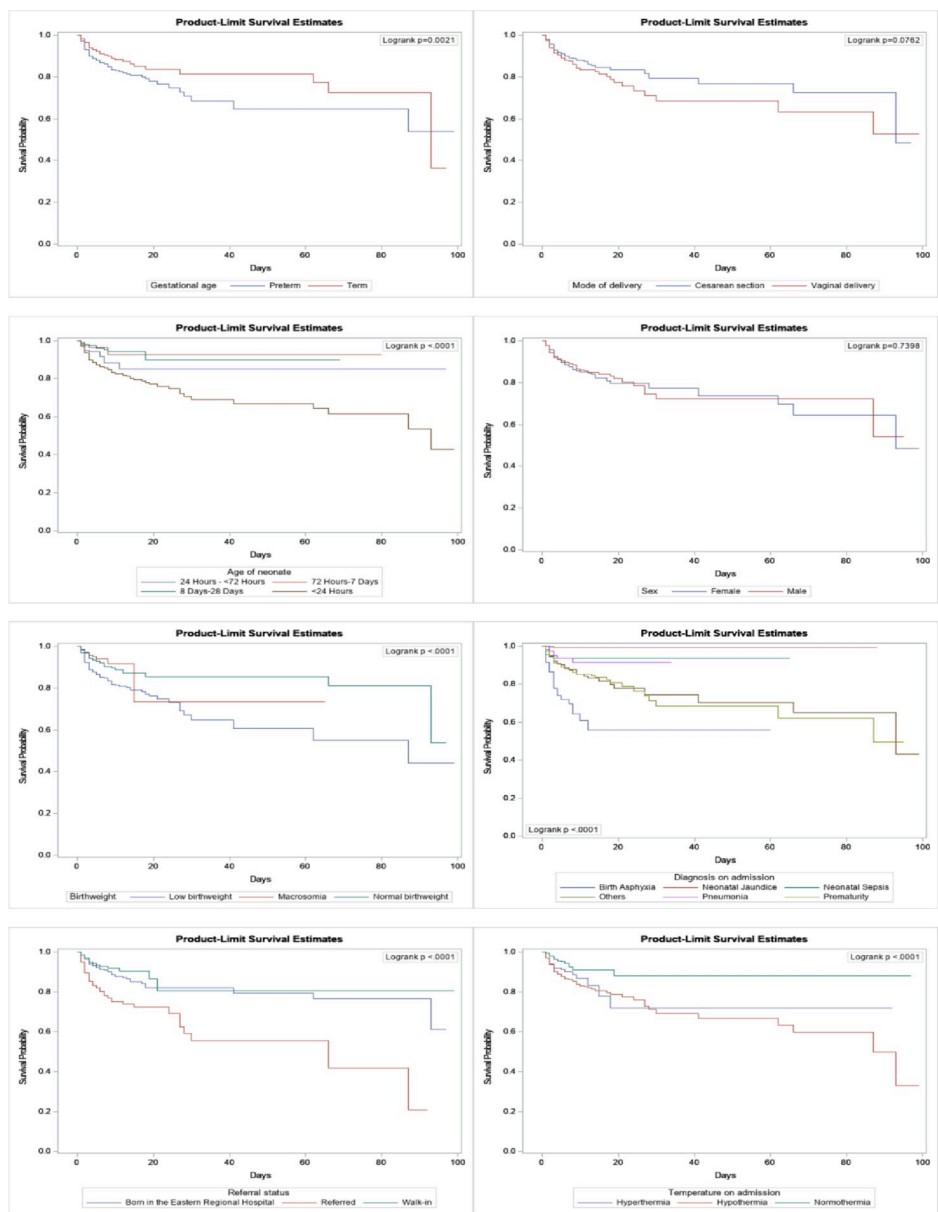

**Fig 2. Kaplan-Meier survival curves of neonates admitted in the NICU of the Eastern Regional Hospital by maternal characteristics, 2022.**

whose mothers were multiparous was 1.66 times the rate among neonates whose mothers were primiparous (Table 2).

## Discussion

We assessed 1684 neonates admitted in the NICU of the Eastern Regional Hospital in Ghana and provides insight on factors associated with neonatal mortality. We observed that low birthweight, hypothermia, hyperthermia, birth asphyxia and multiparity were associated with higher neonatal mortality. On the contrary, age of neonate, referral status and eight or more antenatal contacts were associated with lower mortality.

**Table 2. Predictors of neonatal mortality.**

| Variable | Unadjusted HR (95% CI) | P-value | Adjusted HR (95% CI) | P-value |
|---|---|---|---|---|
| Place of residence | | | | |
| Rural | Reference | | Reference | |
| Urban | 0.63 (0.47,0.83) | 0.0013 | 0.77 (0.57,1.06) | 0.107 |
| Number of antenatal contacts | | | | |
| 0 | Reference | | Reference | |
| 1—7 | 1.55 (1.10,2.19) | 0.0134 | 0.90 (0.59,1.39) | 0.632 |
| ≥8 | 0.87 (0.57,1.33) | 0.5232 | 0.54 (0.32,0.92) | 0.024 |
| Parity | | | | |
| Primiparous | Reference | | Reference | |
| Multiparous | 1.66 (1.15,2.40) | 0.0073 | 1.66 (1.02,2.70) | 0.040 |
| Gestational age | | | | |
| Term | Reference | | Reference | |
| Preterm | 1.55 (1.17,2.06) | 0.0025 | 1.22 (0.76,1.95) | 0.415 |
| Age of neonate | | | | |
| <24 Hours | Reference | | Reference | |
| 24 Hours - <72 Hours | 0.57 (0.34,0.97) | 0.0398 | 0.76 (0.43,1.33) | 0.330 |
| 72 Hours-7 Days | 0.33 (0.14,0.80) | 0.0141 | 0.77 (0.30,1.96) | 0.578 |
| 8 Days-28 Days | 0.29 (0.16,0.53) | <.0001 | 0.41 (0.21,0.81) | 0.011 |
| Birthweight | | | | |
| Normal weight | Reference | | Reference | |
| Low birthweight | 1.93 (1.42,2.62) | <.0001 | 1.63 (1.04,2.54) | 0.032 |
| Macrosomia | 0.84 (0.45,1.55) | 0.5693 | 0.97 (0.48,1.93) | 0.919 |
| Diagnosis on admission | | | | |
| Neonatal Sepsis | Reference | | Reference | |
| Prematurity | 2.09 (1.08,4.05) | 0.0293 | 0.88 (0.42,1.82) | 0.721 |
| Neonatal Jaundice | 0.11 (0.03,0.52) | 0.0049 | 0.12 (0.03,0.55) | 0.006 |
| Birth Asphyxia | 5.55 (2.61,11.79) | <.0001 | 3.69 (1.68,8.11) | 0.001 |
| Pneumonia | 1.08 (0.39,2.97) | 0.8841 | 1.27 (0.46,3.55) | 0.646 |
| Other diseases | 1.98 (1.02,3.87) | 0.0452 | 1.41 (0.71,2.80) | 0.326 |
| Referral status | | | | |
| Referred | Reference | | Reference | |
| Walk-in | 0.33 (0.21,0.53) | <.0001 | 0.54 (0.32,0.90) | 0.017 |
| Born in the Eastern Regional Hospital | 0.41 (0.31,0.55) | <.0001 | 0.39 (0.28,0.55) | <.001 |
| Temperature on admission | | | | |
| Normothermia | Reference | | Reference | |
| Hyperthermia | 2.00 (1.15,3.47) | 0.0141 | 1.85 (1.01,3.39) | 0.048 |
| Hypothermia | 2.49 (1.66,3.73) | <.0001 | 1.82 (1.16,2.85) | 0.009 |
| Mother's age* | 1.01 (0.99,1.02) | 0.3059 | — | |
| Marital status* | | | — | |
| Single | Reference | | | |
| Married | 0.92 (0.69,1.21) | 0.3909 | | |
| Mother's education* | | | — | |
| None | Reference | | | |
| Primary | 0.89 (0.54,1.47) | 0.6415 | | |
| Secondary | 0.85 (0.58,1.24) | 0.4055 | | |
| Tertiary | 0.70 (0.44,1.12) | 0.1357 | | |
| Multiple gestation* | | | — | |

(*Continued*)

**Table 2.** (Continued)

| Variable | Unadjusted HR (95% CI) | P-value | Adjusted HR (95% CI) | P-value |
|---|---|---|---|---|
| No | Reference | | | |
| Yes | 0.88 (0.58,1.35) | 0.5677 | | |
| Mode of delivery* | | | — | |
| Vaginal delivery | Reference | | | |
| Cesarean section | 0.78 (0.59,1.03) | 0.0799 | | |
| Sex of neonate* | | | — | |
| Male | Reference | | | |
| Female | 1.05 (0.79,1.39) | 0.7419 | | |

*Variables were excluded from the adjusted model because they were not significant in the unadjusted model.

We found that low birthweight was associated with higher mortality rate compared to normal birthweight. This could be attributed to several factors. Neonates with low birthweight tend to have a larger surface area with inadequate subcutaneous and brown fat, which are required to maintain normothermia [21, 22]. This makes them prone to hypothermia and are more susceptible to infections [23–25], which are associated with neonatal deaths. The association between low birthweight and neonatal mortality has been reported in previous studies [11, 26, 27]. Consistent with prior studies [28–31], we observed that hypothermia and hyperthermia were independently associated with neonatal mortality. The temperature of a neonate on admission is a survival indicator and therefore maintaining normothermia is crucial to reducing neonatal deaths.

Our study also observed that neonates with birth asphyxia had a higher mortality rate compared to those with neonatal sepsis. Although existing literature have demonstrated that both birth asphyxia and neonatal sepsis are independently associated with neonatal mortality [20, 26, 27, 32], our finding may be a reflection of the severity of birth asphyxia in contrast with the insidious nature of neonatal sepsis. On the other hand, we found that NMR was lower among neonates with neonatal jaundice compared to those with neonatal sepsis. We attributed this finding to the many cases of neonatal sepsis who reported late for treatment at the Eastern Regional Hospital and therefore presented with more serious and life-threatening complications compared to cases of neonatal jaundice, which were often diagnosed early at the hospital after birth.

Neonates born in the Eastern Regional Hospital and neonates whose mothers reported to the Eastern Regional Hospital on a walk-in basis had a lower mortality rate compared to neonates who were referred to the Eastern Regional Hospital. This finding is not surprising as cases referred from other facilities to the Eastern Regional Hospital tend to be severe with complications and require specialist care, which is not available at primary health care facilities. The higher mortality among neonates who were referred to the Eastern Regional Hospital is a wake-up call to strengthen the public health system at the primary healthcare level. This will help reduce mortality among neonates who are referred. Consistent with previous studies, we also found significant association between multiparity and neonatal mortality [33–35].

Furthermore, neonates whose mothers had at least 8 antenatal care contacts had lower mortality compared to neonates whose mothers did not initiate antenatal care. The WHO posits that a minimum of 8 antenatal care contacts are recommended to reduce perinatal mortality and improve women's experience of care [36]. Our finding of a lower NMR among neonates whose mothers made 8 or more antenatal care contacts lends support to the WHO's call on antenatal care for a positive pregnancy experience [36]. We also observed that neonates aged

8–28 days had a lower mortality compared to neonates who were less than 24 hours old as most of the neonatal deaths (9.9%) occurred within the first 24 hours of life. This finding is consistent with several prior studies [37–39]. Effective and high-quality obstetric and early neonatal care within the first 24 hours of life is paramount to reducing neonatal mortality.

The overall incidence of neonatal mortality in our study (11.82%) was different from other hospitals in Ghana. Our finding was higher than that reported in the Upper West Regional Hospital (8.91%), but lower than that observed in the Tamale Teaching Hospital (13.4%), Komfo Anokye Teaching Hospital (20.2%), and the Korle-Bu Teaching Hospital (19.2%) [16–18, 40]. The differences in neonatal mortality could be due to the different level of health-care provision by each facility (i.e., whether it's a secondary or tertiary level of care) [41]. It may also be due to differences in access to specialist staff, medical equipment and treatment options available to these hospitals as well as the severity of neonates admitted [17]. Therefore, the lower mortality observed in our study compared to many other referral hospitals in Ghana could be due to the high skilled staff to neonate ratio at the NICU and medical equipment at their disposal.

Our study had some limitations. This study was limited to the data in the electronic medical records of the hospital and therefore data on other factors, including potential confounders (e.g., timing on onset of symptoms to admission, household wealth, home delivery, etc.) were not available. The NMR could have been overestimated compared to the true rate in the general population as our study was conducted in a secondary referral hospital in the Eastern Region of Ghana. Our findings may not be generalizable, nonetheless our cohort study offers insight on predictors of neonatal mortality, which might be useful for the development of policies and interventions to strengthen neonatal care delivery and antenatal care programs among similar hospital settings in Ghana.

## Conclusions

We found that neonatal characteristics such as low birthweight, hypothermia, hyperthermia, and birth asphyxia were associated with higher rate of neonatal mortality. However, neonates aged 8–28 days, those born in the Eastern Regional Hospital and walk-in mothers with neonates were associated with lower mortality rate. Neonates diagnosed with neonatal jaundice had lower hazard rate of dying compared to neonates with sepsis. Whilst our study also demonstrated that maternal characteristic such as multiparity was associated with a higher NMR, neonates whose mothers had 8 or more antennal contacts had lower NMR. These findings are useful for informed decision making with regards to predictors of neonatal mortality in the Eastern Region. Further studies should consider evaluating the process of neonatal care delivery and the antenatal care program to establish why most mothers did not meet the WHO recommendation of 8 or more antenatal care contacts. It's also essential to strengthen the primary healthcare system to reduce mortality associated with neonates referred to secondary heath facilities.

## Supporting information

**S1 Dataset. SAS7bdat file used for the analysis.**
(SAS7BDAT)

## Author Contributions

**Conceptualization:** Diana Awintima Apanga, Paschal Awingura Apanga.

**Formal analysis:** Paschal Awingura Apanga.

**Methodology:** Diana Awintima Apanga, Maxwell Tii Kumbeni, Abdulai Mohammed Salifu, Nana Mireku-Gyimah, Paschal Awingura Apanga.

**Project administration:** Diana Awintima Apanga.

**Supervision:** Paschal Awingura Apanga.

**Writing – original draft:** Diana Awintima Apanga, Maxwell Tii Kumbeni, Abdulai Mohammed Salifu, Nana Mireku-Gyimah, Paschal Awingura Apanga.

**Writing – review & editing:** Diana Awintima Apanga, Maxwell Tii Kumbeni, Abdulai Mohammed Salifu, Nana Mireku-Gyimah, Paschal Awingura Apanga.

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
