## [Decision Letter · Decision Letter 0]

7 Nov 2023

PGPH-D-23-01616

Predictors of neonatal mortality in the Eastern Regional Hospital in Ghana: A retrospective cohort study

Dear Dr. Apanga,

Thank you for submitting your manuscript to PLOS Global Public Health. After careful consideration, we feel that it has merit but does not fully meet PLOS Global Public Health’s publication criteria as it currently stands. Therefore, we invite you to submit a revised version of the manuscript that addresses the points raised during the review process.

Methods section of the paper requires more details and clarity on the design and accompanying appropriate statistical analysis.

We look forward to receiving your revised manuscript.

Kind regards,

Sonali Sarkar

Academic Editor

Journal Requirements:

Additional Editor Comments (if provided):

Reviewers' comments:

Reviewer's Responses to Questions

**Comments to the Author**

1. Does this manuscript meet PLOS Global Public Health’s publication criteria? Is the manuscript technically sound, and do the data support the conclusions? The manuscript must describe methodologically and ethically rigorous research with conclusions that are appropriately drawn based on the data presented.

Reviewer #1: Partly

Reviewer #2: Yes

2. Has the statistical analysis been performed appropriately and rigorously?

Reviewer #1: Yes

Reviewer #2: Yes

3. Have the authors made all data underlying the findings in their manuscript fully available (please refer to the Data Availability Statement at the start of the manuscript PDF file)?

Reviewer #1: Yes

Reviewer #2: No

4. Is the manuscript presented in an intelligible fashion and written in standard English?

Reviewer #1: Yes

Reviewer #2: Yes

5. Review Comments to the Author

Reviewer #1: 1. The efforts of the team are well appreciated. It is indeed an important topic.

2. While studying the predictors of mortality, rather than categorising the patient as walk in or born at eastern region hospital, it would be more prudent to categorise as born in hospital setting, home setting with no trained personnel, home setting with trained personnel, with or without delivery kits. That would give more information on why did the other group not born in eastern regional hospital has a higher hazards ratio of dying. This will make the conclusion more meaningful to a clinician. If this information is available or can be extracted, it should be added.

3. The authors have reported in discussion that the mortality was lower in neonatal jaundice, and higher in asphyxia than neonatal sepsis. This finding should be read with caution as the study was not designed with the objective to study the mortality rates in different disease conditions and may not be powered enough. It would be better to use the phrase “our study observed” rather than “our study found”. This should also not be a part of conclusion. At the most it can say that which group had the highest hazards of dying and which group had the lowest.

4. The last conclusion, “We recommend the implementation of interventions that would ensure prompt and high-quality neonatal care to avert mortalities from the complications that occur in newborns” cannot be concluded or recommended from the current study as the study neither has defined any intervention, nor it has tested it.

Minor comments

1. Line 137: Majority seems a wrong word to describe 65.1% of population

2. Line 139: Years is missing after 27.1

3. There are minor grammatical errors through the manuscript that need to be corrected.

Reviewer #2: It is a fantastic effort by the author team to look at the predictors of neonatal mortality carefully. The manuscript in its current form is not fully ready for publication. I am attaching some comments for improving this manuscript—all the best and congratulations.

Overall comments

1. Authors may consider adding a time component in the title to give the audience a clear message regarding the study's timeline.

2. The authors used secondary data from the EHR of the hospital belonging to one episode of admission and outcome among the neonates. Therefore, the study design claimed in the paper (retrospective cohort study) is debatable. Since there was no follow-up period and outcomes were during the same episode of admission, the authors may consider the design as a cross-sectional analytical study using the secondary data from EHR rather than a retrospective cohort.

3. All the predictors identified in the study were already known factors and didn't add anything new to the existing literature. The authors can rework the discussion part to bring out any further information added to the scientific literature from this analysis.

4. The paper is predominantly in passive voice. The authors can consider writing in an active voice, which will bring more clarity to the sentences.

Abstract

5. The abstract fails to bring out the rationale of the study. 

6. The abstract's methods part should clearly show that this is a secondary data analysis using the existing EHR data of Hospital X from Y to Z time stating the critical exposure and study outcomes.

7. The authors try to show all the results in the abstract. Instead, they can limit the key findings alone and try to focus more on the methods part.

8. The recommendations written in the conclusion are very broad. The study doesn't show any results in the implementation status of neonatal care in the hospital but only finds the predictors of mortality. Authors can rethink making the recommendations more specific on the lines of "process evaluation of the neonatal care delivery" and the antenatal care program delivery to ensure 100% compliance. Also, the authors can recommend looking at "why the antenatal care program is not able to provide care for x% of the mothers?"

Introduction

1. It appears that there is already an existing program for addressing neonatal mortality. The rationale is not very clear as the study here is trying to look at the factors, which are already established risk factors for neonatal mortality. 

2. The time component is missing in the study's aim.

Methods

3. The authors claim this study to be a retrospective cohort which means that the birth details were extracted from the past timeline and followed them up to quantify their outcomes. However this information is missing in the methods section. What was the frequency of follow-up? Who followed them up? The author team may consider this a cross-sectional analytical study using EHR-based secondary data.

4. Details on how the data were entered into the Electronic Health Record (EHR) need to be provided. 

4. The authors may add details regarding the admission pattern, accessibility of the hospital, number of beds in the NICU, facilities available, including workforce and equipment, etc, to the methods section to provide the context for answering the research question. Is this the highest facility available in the region (this can affect the admission pattern)? Description of the setting is important.

5. The authors claim that the study's primary outcome was time to death. But the primary outcome is the status of the neonate at a specified period – dead or alive. The team may revisit the paragraph on the outcome to bring this out clearly.

6. What software did the study team use for analysis? Kindly provide the same with adequate reference.

7. What is the source of the live births data for calculation of NMR? This should be clearly stated in the methods section.

8. The methods section should specify the calculation of unadjusted HR, and then adjusting using the Cox regression model should be clearly stated. In the current version, the author explains these in the results section. The author may mention all methods, tests used, etc, in the methods section.

Results

9. The results section starts with the line – "a sample of …". Did the study team perform any sampling for this study? From my understanding of the results – they included all the admissions to NICU during the study period (no sample). The team can consider rewriting this line.

10. The percentages written in the parenthesis are confusing. In line 142, the authors say – Among the neonates who died, 73 were premature. (73/199 should be around 36%, but it's written 4.4%). Authors can clearly state the denominators used. Similar issue in line 143 also (167/199??).

11. The author may save words by avoiding mentioning the tests (line 144) used for comparisons in the results section. It's already mentioned in the methods section.

12. In Table 1, the parenthesis value issue exists. Generally, in a cohort study, we try to show row percentages. But here it's all overall percentages. The authors may reconsider this.

13. Kindly consider reducing the p-value to three decimal places and the ink-to-data ratio in all tables.

14. Figures 1 and 2 are not clear in the reviewer copy. Kindly submit high-quality images for publication consideration.

15. Data Quality is an essential component in secondary data analysis. The authors may comment on this in the results section. Was there only 1684 admissions? Were there any duplicates – if yes, how did you handle the same should be explained.

Discussion

16. The discussion section can improve a lot. The first paragraph can be the key findings alone –a maximum of three take-home messages. An in-depth discussion of these critical findings follows them. However, in the current version – the first paragraph is too detailed with comparisons to other studies. The author team may consider cleaning it up to reveal two or three key messages.

17. Also, after making necessary comparisons with existing studies, the author team should be able to interpret the findings for the current settings (conclusion and recommendation for that finding). The current version of the manuscript is deficient in this.

18. In lines 227-230, the team can speak about the type and level of facilities –a significant factor that can affect mortality (the author team should add the details of the current health facility in the methods and discuss in detail in the discussion).

19. In lines 237-238, the author's team discusses using these results for policy and interventions. The team can make it more specific than being broad. For example, the maximum deaths are due to prematurity. In future studies, we can try to understand why premature births occur and then act on those – which could be a more specific recommendation. Also, the team can suggest a program evaluation to understand the quality of care being delivered at the facility.

20. The recommendations written in the conclusion part are very broad. The study doesn't show any results in the implementation status of neonatal care in the hospital but only finds the predictors of mortality. Authors can rethink making the recommendations more specific on the lines of "process evaluation of the neonatal care delivery" and the antenatal care program delivery to ensure 100% compliance. Also, the authors can recommend looking at "why the antenatal care program is not able to provide care for x% of the mothers?"

21. The authors can conclude with what they think as a further study on using this data for decisions. The study team can also propose a qualitative approach to understanding why women don't get antenatal treatment.

6. PLOS authors have the option to publish the peer review history of their article (what does this mean?). If published, this will include your full peer review and any attached files.

**Do you want your identity to be public for this peer review?** For information about this choice, including consent withdrawal, please see our Privacy Policy.

Reviewer #1: No

Reviewer #2: **Yes: **Dr. Sharan Murali

While revising your submission, please upload your figure files to the Preflight Analysis and Conversion Engine (PACE) digital diagnostic tool, https://pacev2.apexcovantage.com/. PACE helps ensure that figures meet PLOS require

---

## [Editor Report · Decision Letter 1]

22 Jan 2024

PGPH-D-23-01616R1

Predictors of neonatal mortality in the Eastern Regional Hospital in Ghana: A retrospective cohort study

Dear Dr. Apanga,

Thank you for submitting your manuscript to PLOS Global Public Health. After careful consideration, we feel that it has merit but does not fully meet PLOS Global Public Health’s publication criteria as it currently stands. Therefore, we invite you to submit a revised version of the manuscript that addresses the points raised during the review process.

Responses to the reviewers comments are well accepted. Modification in the language and additions done as per the suggestions of the reviewers is appreciated. However, a few more aspects, including the justification for the very high risk group in the study, can further improve the paper.

We look forward to receiving your revised manuscript.

Kind regards,

Sonali Sarkar

Academic Editor

Journal Requirements:

Additional Editor Comments:

Thank you for modifying the paper. Few additional suggestions are as follows.

1. Line no 65, 66 - the word steady means decline is good. I think the authors mean that the decline is slow rather than steady.

2. This study is conducted on the very high risk category on neonates admitted to the NICU. Justification for the inclusion criteria needs to be included in the introduction.

3. Was sample size calculation attempted? It would be good to state the calculated sample size to validate that the study was powered to identify the associations identified.

4. Discussion can focus more on the higher mortality among the referred neonates, adding the existing level of care at the primary level of public health system and the importance of strengthening it in the recommendations.

---

## [Decision Letter · Decision Letter 2]

10 May 2024

Predictors of neonatal mortality in the Eastern Regional Hospital in Ghana: A retrospective cohort study

PGPH-D-23-01616R2

Dear Dr Apanga,

We are pleased to inform you that your manuscript 'Predictors of neonatal mortality in the Eastern Regional Hospital in Ghana: A retrospective cohort study' has been provisionally accepted for publication in PLOS Global Public Health.

Best regards,

Julia Robinson

Executive Editor

Reviewer Comments (if any, and for reference):

Reviewer's Responses to Questions

**Comments to the Author**

1. If the authors have adequately addressed your comments raised in a previous round of review and you feel that this manuscript is now acceptable for publication, you may indicate that here to bypass the “Comments to the Author” section, enter your conflict of interest statement in the “Confidential to Editor” section, and submit your "Accept" recommendation.

Reviewer #1: All comments have been addressed

2. Does this manuscript meet PLOS Global Public Health’s publication criteria? Is the manuscript technically sound, and do the data support the conclusions? The manuscript must describe methodologically and ethically rigorous research with conclusions that are appropriately drawn based on the data presented.

Reviewer #1: Yes

3. Has the statistical analysis been performed appropriately and rigorously?

Reviewer #1: Yes

4. Have the authors made all data underlying the findings in their manuscript fully available (please refer to the Data Availability Statement at the start of the manuscript PDF file)?

Reviewer #1: Yes

5. Is the manuscript presented in an intelligible fashion and written in standard English?

Reviewer #1: Yes

6. Review Comments to the Author

Reviewer #1: The manuscript seems fine now. Though i believe that the statement "It’s also essential to strengthen the primary healthcare system to reduce mortality associated with neonates referred to secondary heath facilities" is not a conclusion of the study, and is rather a general statement.

7. PLOS authors have the option to publish the peer review history of their article (what does this mean?). If published, this will include your full peer review and any attached files.

**Do you want your identity to be public for this peer review?** For information about this choice, including consent withdrawal, please see our Privacy Policy.

Reviewer #1: **Yes: **Dr Gunjan Kumar
